# Omacor Protects Normotensive and Hypertensive Rats Exposed to Continuous Light from Increased Risk to Malignant Cardiac Arrhythmias

**DOI:** 10.3390/md19120659

**Published:** 2021-11-24

**Authors:** Tamara Egan Benova, Csilla Viczenczova, Barbara Szeiffova Bacova, Jitka Zurmanova, Vladimir Knezl, Katarina Andelova, Narcis Tribulova

**Affiliations:** 1Center of Experimental Medicine, Slovak Academy of Sciences, 84104 Bratislava, Slovakia; tamara.benova@savba.sk (T.E.B.); vladimir.knezl@savba.sk (V.K.); katarina.andelova@savba.sk (K.A.); 2Research Center for Molecular Medicine of the Austrian Academy of Sciences, A-1090 Vienna, Austria; viczencz.csilla@gmail.com; 3Department of Physiology, Faculty of Science, Charles University, 50003 Hradec Kralove, Czech Republic; jitka.zurmanova@natur.cuni.cz

**Keywords:** omacor, light pollution, rats, cardiac arrhythmias, connexin-43, NF-κB, iNOS

## Abstract

Light pollution disturbs circadian rhythm, and this can also be deleterious to the heart by increased susceptibility to arrhythmias. Herein, we investigated if rats exposed to continuous light had altered myocardial gene transcripts and/or protein expression which affects arrhythmogenesis. We then assessed if Omacor^®^ supplementation benefitted affected rats. Male and female spontaneously hypertensive (SHR) and normotensive Wistar rats (WR) were housed under standard 12 h/12 h light/dark cycles or exposed to 6-weeks continuous 300 lux light for 24 h. Half the rats were then treated with 200 mg/100 g b.w. Omacor^®^. Continuous light resulted in higher male rat vulnerability to malignant ventricular fibrillation (VF). This was linked with myocardial connexin-43 (Cx43) down-regulation and deteriorated intercellular electrical coupling, due in part to increased pro-inflammatory NF-κB and iNOS transcripts and decreased sarcoplasmic reticulum Ca^2+^ATPase transcripts. Omacor^®^ treatment increased the electrical threshold to induce the VF linked with amelioration of myocardial Cx43 mRNA and Cx43 protein levels and the suppression of NF-κB and iNOS. This indicates that rat exposure to continuous light results in deleterious cardiac alterations jeopardizing intercellular Cx43 channel-mediated electrical communication, thereby increasing the risk of malignant arrhythmias. The adverse effects were attenuated by treatment with Omacor^®^, thus supporting its potential benefit and the relevance of monitoring omega-3 index in human populations at risk.

## 1. Introduction

Light pollutions and continuous light associated with melatonin deficiency disturb circadian rhythm [1,2], and constant light hampers pineal gland function and abolishes approximately 90% of nocturnal melatonin production [3]. In addition, circadian disorders are deleterious to the heart and may enhance susceptibility to cardiac arrhythmias [4,5].

Rats are photo-periodic animals which favour darkness over light. They are sensitive to light intensities as low as 60 lux, and while this is sufficient to cause distress [6], prolonged exposure to light intensities above 200 lux is significantly deleterious [7]. Chronic light pollution has also been reported to have a strong link to metabolic diseases [8], where continuous light promotes increased systolic blood pressure, myocardial structural remodelling [9,10] and increases the heart’s vulnerability to arrhythmias [11].

Myocardial connexin-43 (Cx43) is important in the development of malignant cardiac arrhythmias [12] because it is responsible for electrical communication and action potential transmission through the Cx43 channels directly linking neighbouring cardiomyocytes. Previous studies [13,14,15] support our finding that Cx43 down-regulation and its abnormal topology jeopardize myocardial electrical function and stability. Noteworthy, the pro-inflammatory factors, such as NF-κB (the nuclear factor kappa-light-chain-enhancer of activated B cells) and iNOS (inducible nitric oxide synthase) have important impact on this process [15,16,17]. While abnormal calcium handling and calcium overload due to disorders in SERCA2 (sarcoplasmic reticulum Ca^2+^ATPase) promotes cell-to-cell electrical uncoupling at the Cx43 channels as well as ectopic action potential generation [15,18].

However, interventions which prevent and/or attenuate Cx43 disorders can provide benefits [14,19], and it is important here that the unique properties of omega-3 polyunsaturated fatty acids, such as Omacor^®^ are permanent subjects of investigation in the protection of cardiovascular health and prevention of cardiac arrhythmias [20]. In this context it should be emphasized that Omacor^®^ used in this study exerts relevance to marine biosciences due to a marine-originated, dedicated plant that is a source of the high content of eicosapentanoic acid (EPA) and docosahexanoic acid (DHA). Fish do not produce omega-3 fatty acids by themselves but obtain it by consuming the original source in their diets—marine microalgae, also known as marine phytoplankton. Thus, phytoplankton as a fish food (fish-based product) is a source of 460 mg EPA and 380 mg DHA in the form of ethyl ester per capsule of Omacor^®^ of Pronova, BioPharma [21] that was used in current study. We have reported the cardiovascular benefit of Omacor^®^ manufactured by this leading company in the research and development of marine-originated omega-3 fatty acids in the Norway in numerous previous studies [20,22,23] as well as recent studies [24]. Omacor^®^ has also been approved for the secondary prevention of myocardial infarction. Of note, production capability for marine formula Omacor^®^ is rapidly increasing due to efficient treatment not only of cardiovascular disease but also non-communicable diseases like cancer, anxiety, depression, and Alzheimer’s disease. The drug of marine origin has shown in a several of clinical trials to be a potent anti-inflammatory and antiarrhythmic molecule [20]. Undoubtedly, the development and production of biologically and therapeutically active compounds from the sea is actual task even in the era of dominant synthetic drugs production.

We examined if the exposure of hypertensive and normotensive rats to continuous light affects myocardial Cx43 and the heart’s vulnerability to malignant arrhythmias. We tested our hypothesis that supplementing rats with marine drug formula Omacor^®^ is cardioprotective and preserves Cx43 in ‘light smog’.

## 2. Results

### 2.1. Effects of Treatment with Omacor^®^ on Rat Systolic Blood Pressure and Biometric Parameters of Rats

The registered biometric parameters are summarized in Table 1. Systolic blood pressure (SBP) was increased in both male and female SHR rats but not in the WR. Both WR and SHR rats exposed to continuous light for 6 weeks had significant SBP increase, and this was less pronounced in the females. Omacor^®^ treatment significantly suppressed the SBP in both male strains and there also some benefit for the female rats. Neither continuous light nor Omacor^®^ affected the absolute or relative heart weight (HW) and weight of the heart left ventricle (LVW).

### 2.2. Effects of Treatment with Omacor^®^ on Rat Heart Susceptibility to Electrically-Induced Sustained VF

Figure 1A shows that the threshold to induce sustained ventricular fibrillation (VF) was significantly lower in male SHR than in WR hearts, and this difference was less pronounced in females (Figure 1B). The 6-week rat exposure to continuous light resulted in decreased VF threshold significantly in both the WR and SHR male rats but not in the females. Treatment with Omacor^®^ increased the threshold for electrical induction of sustained VF in males regardless of rat strain and the tendency to increase was observed in female SHR as well.

### 2.3. Effects of Treatment with Omacor^®^ on Myocardial Cx43 Gene Transcripts, Protein Levels, and Phosphorylation Status

Examination of GJA1 gene transcripts revealed that exposure of male and female WR and SHR rats to 6-week continuous light resulted in a significant myocardial mRNA decrease for Cx43 compared to rats in standard conditions (Figure 2). Treatment with Omacor^®^ significantly attenuated this unfavourable decrease in male hearts and those of SHR females, but did not decrease GJA1 gene transcripts in WR hearts. 

Besides alterations in Cx43 gene transcript, there were alterations in Cx43 protein abundance (Figure 3) including its functional phosphorylated forms (P1, referred to m.w. 42 kD and P2, referred to m.w. 43 kD) and P0 (m.w. referred to 40 kD) represent non-phosphorylated form of Cx43. Figure 3A highlights that immunoblot analysis showed a significant decrease in myocardial Cx43 protein levels in the hearts of male SHR rats exposed to 6-week continuous light compared to SHR rats in standard light conditions. Surprisingly, the female WR and SHR rats responded to continuous light by increased myocardial Cx43 protein levels compared to those in standard light conditions (Figure 3B). This increase was significant in WR rat hearts. Omacor^®^ ameliorated myocardial Cx43 protein abundance along with its functional phosphorylated status in male SHR and to lesser extent in WR (Figure 3E). Moreover, treatment with Omacor^®^ maintained Cx43 protein levels and its phosphorylated forms comparable to healthy WR in females (Figure 3F). The heart exhibiting reduced Cx43 expression is more susceptible to develop malignant cardiac arrhythmias, while preservation of Cx43 is beneficial to maintain normal rhythm and heart function.

### 2.4. Effect of Treatment with Omacor^®^ on Myocardial PKCε and PKCδ Protein Levels

Phosphorylated Cx43 protein is required for its assembly in channels, and also for the modulation of channel function [15]. PKCε is one of the protein kinases which phosphorylates Cx43 in the heart, and our immunoblotting analysis revealed that its myocardial protein levels were significantly reduced in both male and female SHR rats compared to healthy WR (Figure 4). Unexpectedly, there was an increase in PKCε protein levels after the 6-weeks continuous light exposure in male rats of both strains as well as in females SHR. Omacor^®^ treatment further enhanced the myocardial PKCε protein levels in the SHR strain compared to the levels in basal conditions and continuous light.

PKCδ is known as pro-hypertrophic, pro-apoptotic, and pro-fibrotic myocardial protein [24], whereby such structural remodelling predisposes to malignant cardiac arrhythmias [12,24]. The amount of the PKCδ was significantly increased in both SHR male and female hypertrophic rat hearts (see Table 1) compared to healthy WR heart (Figure 5). The 6-week exposure to continuous light resulted in increased PKCδ protein levels in SHR male rats, but not in the WR males and females. Treatment with Omacor^®^ then suppressed this adverse increase in male SHR rats and decreased the levels in the SHR female.

### 2.5. Effect of Treatment with Omacor^®^ on NF-κB, iNOS and SERCA Gene Transripts

In order to test the anti-inflammatory effects of Omacor^®^, we examined the gene transcript of the kappa-light-chain-enhancing nuclear factor for activated B cells (NF-κB) which is a central inflammatory mediator down-regulating Cx43 [16], thereby promoting occurrence of malignant cardiac arrhythmias. The NF-κB mRNA level was significantly higher in the male SHR rat, but not in the female, compared to WR (Figure 6). The 6-week exposure of rats to continuous light induced pronounced increase in NF-kB gene transcripts in both male and female SHR but not in the WR strain. Omacor^®^ treatment then suppressed the NF-κB mRNA elevation in both male and female SHR rats to the value detected in healthy WR.

In conjunction with NF-κB assessment, we also examined inducible nitric oxide synthase (iNOS) transcripts which can be induced by cytokines thereby impacting myocardial Cx43. Indeed, we have previously found that aberrant iNOS induction contributes to Cx43 degradation that is pro-arrhythmic [17], and the iNOS transcript level was higher in male and female SHR rats compared to the WR strain (Figure 7). Exposure of rats to continuous light for 6 weeks resulted in increased iNOS transcripts in the male SHR rats, but not in the females. Omacor^®^ treatment then suppressed iNOS transcripts in both the SHR males and females.

Ca^2+^-ATPase present in the sarcoplasmic reticulum (SERCA) is the key enzyme involved in Ca^2+^ homeostasis. This is a further important factor affecting the rat’s heart susceptibility to arrhythmias. SERCA disorders which result in intracellular Ca^2+^ overload is highly pro-arrhythmic. Mostly, due to impairment of myocardial communication ensured by Cx43 channels and occurrence of abnormal ectopic electrical activity [18,25]. SERCA mRNA expression level in the normotensive male and female rat was significantly reduced by the 6-weeks continuous light but this was not noted in the SHR strain (Figure 8). Omacor^®^ treatment significantly attenuated this decrease in mRNA SERCA transcript in the male rat and normalised its level in the female WR.

## 3. Discussion

Our results indicate that Omacor^®^ supplementation protects normotensive and hypertensive rats from adverse cardiac alterations when they are subjected to continuous light. This includes preventions of down-regulation of myocardial Cx43 which promotes the development of malignant arrhythmias. Omacor^®^ treatment also suppressed high blood pressure in both the spontaneously hypertensive male and female rats and prevented increased blood pressure (Table 1) from continuous light in the normotensive Wistar rats.

Light pollution associated with circadian rhythm disorders affects cardiac rhythm and can increase susceptibility to arrhythmias [4,5]. Findings herein indicate that both normotensive and hypertensive rats’ exposure to continuous light for 6 weeks resulted in significant reduction in electrical threshold to induce VF (Figure 1) compared to standard light conditions. It is interesting that higher susceptibility to VF was registered in male rats than in females, thus suggesting gender differences. This difference can be explained in part by the higher female rats’ myocardial Cx43 protein expression [26] and the inverse relationship to miR-1 which regulates Cx gene transcript [27]. The downregulation of myocardial Cx43 mRNA (Figure 2) and Cx43 protein (Figure 3) due to continuous light was also much less pronounced in female rats, where normotensive females exhibited even increased myocardial Cx43 protein levels and also its functional phosphorylated forms in response to continuous light. Finally, the effect of Omacor^®^ supplementation to increase VF threshold was also pronounced in both WR and SHR males. Since Omacor^®^ is composed from omega-3 (EPA and DHA) fatty acids, investigation is necessary whether higher endogenous omega-3 fatty acids levels in females than males account for cardiac rhythm protection. These results would also be beneficial for humans, because the omega-3 status of women in Western countries is currently low [28]. Regarding this issue, it should be noted that the plasma levels of omega-3 was not assessed in the current study; however, we have previously reported that plasma EPA and DHA as well as omega-3 index were lower in male and female SHR versus WR but increased after omega-3 intake [29].

In the context of cardiac arrhythmias, it must be emphasized that Cx43 channels are crucial for electrical impulse propagation between cardiomyocytes and synchronized cardiac contraction, while Cx43 down-regulation is highly pro-arrhythmic due to conduction disorders [12,13,14,15]. This down-regulation associated with suppression of its phosphorylated status hampers Cx43 assembly into channels and channel function [15], and the modulation of Cx43 channel function by the PKCε protein kinase may affect the heart’s susceptibility to arrhythmias [30].

Our study revealed an increase of myocardial PKCε protein expression in rats exposed to continuous light and this can be enhanced by Omacor^®^ application (Figure 4), which increased phosphorylated status of Cx43 as well (Figure 3A, E). The Omacor^®^ treated rats have lower susceptibility to ventricular fibrillation and rat hearts may benefit from up-regulation of both Cx43 and PKCε in the stressful condition of continuous light, and this up-regulation may also be advantageous to diabetic and hypothyroid rats [31]. This Omacor^®^ treatment suppressed elevation of myocardial protein abundance of pro-hypertrophic, pro-apoptotic and pro-fibrotic PKCδ myocardial protein in the spontaneously hypertensive rats (SHR, Figure 5). Moreover, these results encourage Omacor^®^ use against hypertension-related myocardial structural remodeling (fibrosis), as it was previously discussed [32].

It has also been reported that the NF-kB acts as a central inflammatory mediator, down-regulating Cx43 [16], and myocardial NF-kB transcripts were enhanced in SHR rats exposed to continuous light. Here, Omacor^®^ apparently suppressed these transcripts in male and female SHR (Figure 6). The inhibited translocation of NF-κB preserved Cx43 channel function [16], and this is consistent with the accepted anti-inflammatory effects of omega-3 fatty acids [20].

Stressful and pro-inflammatory conditions are associated with elevated iNOS, and its aberrant induction contributes to Cx43 degradation [17]. Herein, we identified that rat exposure to continuous light increased iNOS transcript levels in the male SHR rats but not in females SHR, and this may contribute to the lower female cardiac vulnerability to VF. Moreover, treatment with Omacor^®^ suppressed iNOS transcripts in both male and female SHR (Figure 7), and this coincided with Cx43 preservation and decreased VF vulnerability.

In addition, SERCA further affects Cx43 channel function and subsequent cardiac susceptibility to arrhythmias, because it is a key enzyme in intracellular Ca^2+^ homeostasis. Therefore, SERCA disorders and/or down-regulation that result in intracellular Ca^2+^ overload are highly pro-arrhythmic [25] due to Cx43 channel uncoupling and dysfunction [15]. Herein, the myocardial expression level of mRNA SERCA was reduced in normotensive WR rats but not in the SHR strain exposed to continuous light, but Omacor^®^ treatment significantly attenuated this decreased mRNA SERCA in the WR rats (Figure 8).

In summary, our results indicate that long-term exposure of rats to continuous light jeopardizes cardiac Cx43 channels-mediated electrical communication, and this increases the risk of malignant arrhythmia development. In contrast, Omacor^®^ application attenuated the adverse cardiac alterations that promote arrhythmias and preserved the myocardial Cx43 which is essential for the heart’s synchronized electrical function. Besides, our findings support the relevance of omega-3 index monitoring or at least basal blood plasma omega-3 status prior omega-3 fatty acids supplementation. This is important in risky human populations put at risk by an unhealthy lifestyle or persistent light smog exposure. 

In fact, neither blood plasma levels nor omega-3 index were registered in most recent large-scale clinical trial assessing the occurrence of AF [33]. Considering that very heterogenous population of older (>50 years’ males and females) participants received 840 mg/d EPA-DHA (one capsule), it is not surprising that there was not significant difference in the risk of incident AF over five years of follow-up. As we discussed this issue previously [20] the major flaw of clinical studies is lack of information about basal omega-3 status of participants. Accordingly, the relevance of the conclusions of clinical trials is difficult to justify.

## 4. Materials and Methods

### 4.1. Animals and Experimental Design

Experimental procedure: The experiments were conducted in accordance with dictates of the State Veterinary Administration of the Slovak Republic, legislation No. 377/2012, and conform to the European Convention for the Protection of Vertebrate Animals used for Scientific Purposes (1985, Council of Europe No. 123, Strasbourg) and Care and Use of Laboratory Animals (ILAR 1996). All procedures were approved by the Institutional Animal Care Committee. The temperature in animal facilities was between 20–24 °C and humidity between 45–65%, with free access to tap water and standard laboratory nutrition.

Two strains of adult 4-week-old male and female rats were used: normotensive Wistar (WR) and spontaneously hypertensive (SHR). According to the animal care rules three rats were housed in one cage of size 590 mm × 380 mm × 200 mm. Some animals were housed under standard light conditions of 12-hour light phase/12-hour dark phase, and the remainder were exposed to 6-weeks continuous 300 lux light intensity. Those housed in continuous light were treated orally with Omacor^®^ (each rat received directly small piece of biscuit soaked with exact dose of Omacor) for the experiment and compared with non-treated ones. (Omacor^®^: 460 mg EPA and 380 mg DHA per capsule, Pronova, BioPharma Norge AS, Norway, 200 mg EPA+DHA/100 g b.w.) This company use marine phytoplankton as a source of EPA and DHA in the form of ethyl ester per capsule of Omacor^®^ [21]. 

The animals were divided into six groups of males and females (n = 12 per each group). These groups comprised: 1/WR—normotensive rats; 2/WR-L—normotensive rats exposed to constant light; 3/WR-LO—normotensive rats exposed to constant light and treated with Omacor^®^; 4/SHR—hypertensive rats; 5/SHR-L—hypertensive rats exposed to constant light, 6/SHR-LO—hypertensive rats exposed to constant light and treated with Omacor^®^. 

Systolic blood pressure (SBP) was measured non-invasively at the end of the experiment by tail-cuff plethysmography and the Statham Pressure Transducer P23XL (Hugo Sachs, Germany) and the body weight was registered. Rats were euthanized by injection of ketamine (Narketan, Vetoquinol UK Ltd., Towcester, UK, 100 mg/kg). Thoracotomy was followed by heart excision and rinsing the heart in ice-cold physiological saline to halt beating. Heart weight (HW) and left ventricular weight (LVW) were then registered, and the tissue was frozen in liquid nitrogen and stored at −80 °C for further analyses. Finally, six hearts in each group were tested for susceptibility to sustained electrically induced ventricular fibrillation (VF).

### 4.2. Examination of Vulnerability of the Heart to VF

The ex vivo Langendorff-mode perfused heart technique was used to test VF vulnerability, as described in [32,34]. Briefly, the heart was perfused via cannulated aorta with oxygenated Krebs–Henseleit solution (mmol/L: 120 NaCl; 4.2 KCl; 1.75 CaCl_2_; 1.25 MgSO_4_·4H_2_O; 12.5 glucose; 25.0 NaHCO3, pH = 7.4) at constant 80 mmHg pressure (1 mmHg = 133.3 Pa) and 37 °C temperature. A 20-minute period of heart perfusion enabled equilibration of basal hemodynamic parameters, and VF inducibility was then tested by electrical stimulation by electrodes located at epicardia of the right atria and ventricle. A one second burst of electrical rectangular pulses at 15 mA current was delivered by Electrostimulator ST-3 (Medicor, Illatos, Hungary). Finally, stimulus intensity was increased in 5 mA steps to a maximum of 50 mA to determine the threshold if a sustained 2 min 15 mA burst did not induce VF.

### 4.3. Real-Time PCR for mRNA Expression of Cx43, NF-kB, iNOS, PPARy and SERCA

The TRIsure reagent Sigma-Aldrich/(Bioline, Memphis, TN, USA) was used for RNA isolation, and its concentration was determined by NanoDrop ND1000 spectrophotometer (NanoDrop Technologies, Inc., Wilmington, DE, USA). Reverse transcription was performed as previously reported [34] by the RevertAid H Minus first strand cDNA synthesis kit (Fermentas, Hanover, MA, USA). This was achieved with 1.2–1.5 μg of total RNA and random hexamer primer. The resultant single-chain DNA was applied for real-time PCR. Amplification performed on a 7500 fast Real-Time PCR system (Applied Biosystems, Waltham, MA, USA) with 10 μL of SYBR Green PCR master mix containing 30 pmol/L of each primer. Target gene amplification and hypoxantin-guanin phosphoribosyltransferase, housekeeping gene (HRPT)) employed the following primers: for gap junction protein connexin43 (GJA1): 5′-TCCTTGGTGTCTCTCGCTTT-3′(sense) and 5′-GAGCAGCCATTGAAGTAGGC-3′ (antisense); for nuclear factor-κB (N-kB): GAA-TTC-AGC-CCC-TCC-ATT-G (sense) and CTG-AAG-CCT-CGC-TGT-TTA-GG (antisense); for inducible nitric oxide synthase (iNOS): ACC-ATG-GAG-CAT-CCC-AAG-TA (sense) and CAG-CGC-ATA-CCA-CTT-CAG-C (antisense); for sarcoplasmic reticulum Ca^2+^-ATPase (SERCA): ACC-TGG-AAG-ATT-CTG-CGA-AC (sense) and AAT-CCT-GGG-AGG-GTC-CAG (antisense); for HPRT: GAC-CGG-TTC-TGT-CAT-GTC-G (sense) and ACC-TGG-TTC-ATC-ATC-ACT-AAT-CAC (antisense).

The programme comprised an initial AmpliTaq GoldR DNA polymerase activation step at 95 °C for 10 min followed by 50 cycles of denaturation at 95 °C for 15 s, annealing, and elongation at 56 °C for 60 s. Specificity control and a dissociation stage were conducted by sequential temperature increase from 56 to 99 °C, thus recording decreased double-stranded DNA–SYBR Green complex fluorescence strength. Calculations were then performed by the 7500 fast system SDS software (Applied Biosystems, Waltham, MA, USA). The cycle threshold was determined by the number of cycles required for the fluorescence signal to exceed the detection threshold, and the Calculation of the expression of the target gene relative to the housekeeping gene was determined by the difference between the threshold values of the two genes.

### 4.4. SDS-PAGE and Western Blot Analysis of Cx43, PKCε, and PKCδ Protein Levels

Frozen left ventricular tissue was homogenised in lysis buffer of 20% SDS, 10 mmol/L EDTA, 100 mmol/L Tris and pH 6.8. This was diluted in Laemmli buffer. Equal amounts of protein per lane were separated in 10% SDS-polyacrylamide gels at 120 V (Mini-Protean Tetra Cell, Bio-Rad, Hercules, CA, USA). The proteins were transferred to a nitrocellulose membrane (0.2 m pore size, Advantec, Tokyo, Japan) and blocked for 4 h with 5% low-fat milk prior to overnight incubation at 4 °C with the following primary antibodies; anti-Cx43, C6219 Sigma-Aldrich, Saint-Louis, MO, USA; anti-PKCε, sc-214; anti-PKCδ, sc-213; anti-GAPDH, sc-25778—all from Santa Cruz Biotechnology, Inc., Dallas, TX, USA. The membranes were washed with Tris-buffered 0.1% Tween (TBS-T) and incubated for 1 h with a horseradish peroxidase-linked secondary antibody (anti-rabbit, diluted 1:2000, Cell Signalling Technology, Danvers, MA, USA, #7074), then washed in TBS-T and visualised by enhanced chemiluminescence. Finally, the Protein bands were quantified by version 5.0 Carestream molecular imaging software densitometric analysis (Carestream Health, New Haven, CT, USA) and normalisation to the GAPDH band. The representative immunoblots (proteins and GAPDH) in figures were spliced to allow better comparison. The spliced blots were from the same gel.

### 4.5. Statistical Analysis

The Kolmogorov–Smirnov normality test assessed if the variables are normally distributed. One-way ANOVA and Tukey’s multiple comparison tests then estimated differences between groups, and data was expressed as mean + SD with *p* < 0.05 statistically significant.

## 5. Conclusions

Our results indicate that light pollution disturbs circadian rhythm, and exposure of rats to continuous light can be deleterious to the heart by increasing its susceptibility to malignant cardiac arrhythmias. Noteworthy, subsequent treatment with Omacor^®^ increased the electrical threshold required to induce life-threatening ventricular fibrillation. This was linked with amelioration of Cx43 mRNA and Cx43 myocardial protein levels and the suppression of NF-kB and iNOS transcripts elevation. Findings of the current study strongly imply the antiarrhythmic properties of omega-3 (EPA and DHA) fatty acids in condition of continuous light exposure that increase susceptibility of the heart to develop arrhythmias.

## Figures and Tables

**Figure 1 marinedrugs-19-00659-f001:**
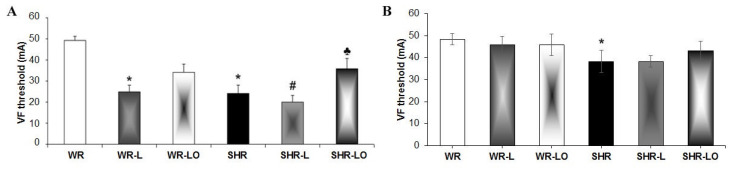
The threshold to induce sustained ventricular fibrillation (VF) is lower in male (**A**) and female (**B**) spontaneously hypertensive rat (SHR) hearts than normotensive rats (WR). The VF threshold was significantly reduced after 6-week continuous light in WR and SHR males but not in females of either strain. Treatment with Omacor^®^ increased the electrical threshold for induced sustained VF in males of both rat strains. The values here are the mean ± SD of 6 rats in each group. * *p* < 0.05 compared to WR; # *p* < 0.05 compared to SHR; ♣ *p* < 0.05 compared to SHR-L.

**Figure 2 marinedrugs-19-00659-f002:**
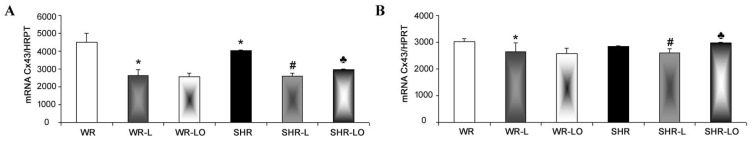
Myocardial Cx43 mRNA transcript is lower in male SHR rats than the WR strain (**A**), but no difference was detected in females (**B**). There is decreased Cx43 mRNA after 6-week continuous light in the male and female WR and SHR rats. Omacor^®^ attenuated this decrease in SHR but did not affect it in the WR strain. Values are the mean ± SD of 6 rats in each group. * *p* < 0.05 compared to WR; # *p* < 0.05 compared to SHR; ♣ *p* < 0.05 compared to SHR-L.

**Figure 3 marinedrugs-19-00659-f003:**
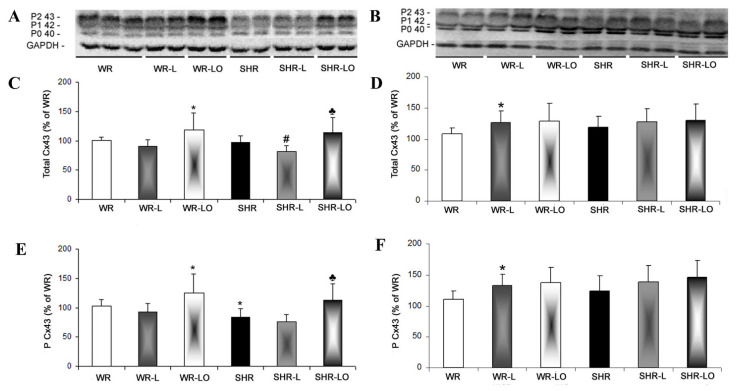
Immunoblot demonstration (**A**,**B**) of myocardial Cx43 protein bands of functional P1 and P2 phosphorylated forms and the PO non-phosphorylated form in experimental rats. There is a tendency to decreased Cx43 levels in male WR and SHR rats from 6-week continuous light compared to rats in standard light conditions (**C**). Surprisingly, female WR responded to continuous light by increased Cx43 protein levels compared to rats in standard light conditions (**D**). Omacor^®^ ameliorated myocardial Cx43 protein levels and functional phosphorylated status in the WR and SHR males (**E**) and maintained in females (**F**). Values are the mean ± SD of 6 rats in each group. * *p* < 0.05 compared to WR; # *p* < 0.05 compared to SHR; ♣ *p* < 0.05 compared to SHR-L.

**Figure 4 marinedrugs-19-00659-f004:**
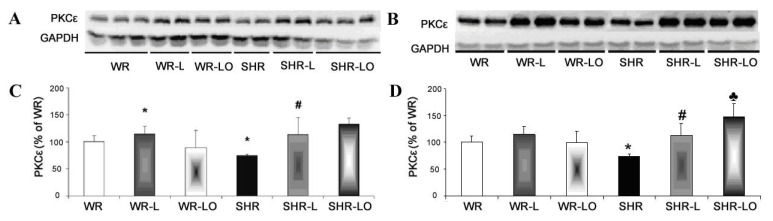
Myocardial PKCε protein levels are significantly reduced in SHR male (**A**,**C**) and female (**B**,**D**) rats compared to their WR counterparts. There was increased PKCε in response to 6-week continuous light regardless of rat strain or gender. Omacor^®^ enhanced myocardial PKCε protein levels in SHR rats compared to the levels under basal or continuous light conditions. Values are the mean ± SD of 6 rats in each group. * *p* < 0.05 compared to WR; # *p* < 0.05 compared to SHR; ♣ *p* < 0.05 compared to SHR-L.

**Figure 5 marinedrugs-19-00659-f005:**
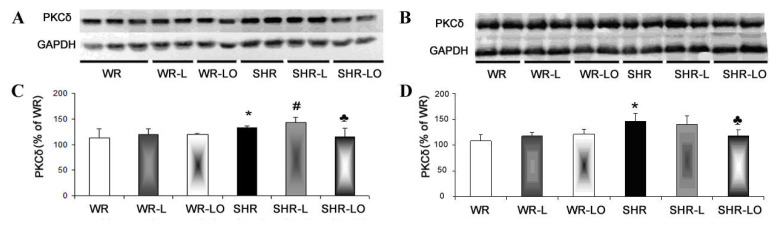
PKCδ is increased in both male (**A**,**C**) and female (**B**,**D**) spontaneously hypertensive rat hearts (SHR) compared to normotensive rats (WR). The 6-week exposure of animals to continuous light resulted in increased PKCδ protein levels in SHR male rats, but not in the WR males and females. Omacor^®^ suppressed PKCδ protein levels in both SHR males and females. Values are the mean ± SD of 6 rats in each group. * *p* < 0.05 compared to WR; # *p* < 0.05 compared to SHR; ♣ *p* < 0.05 compared to SHR-L.

**Figure 6 marinedrugs-19-00659-f006:**
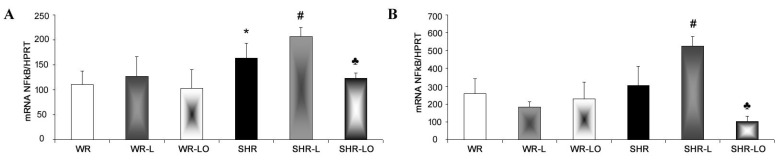
The NF-κB mRNA expression level is higher in male (**A**) spontaneously hypertensive rat (SHR), but not in the female (**B**), compared to normotensive rats (WR). Exposure of rats to 6-week continuous light induced pronounced of NF-κB transcript increase in the male and female SHR but not in WR. Omacor^®^ did suppress elevation of NF-κB mRNA elevation in both the male and female SHR to the value detected in healthy WR rats. Values are the mean ± SD of 6 rats in each group. * *p* < 0.05 compared to WR; # *p* < 0.05 compared to SHR; ♣ *p* < 0.05 compared to SHR-L.

**Figure 7 marinedrugs-19-00659-f007:**
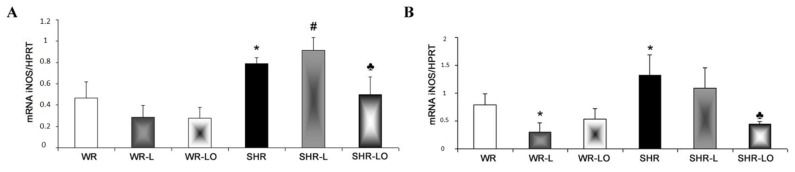
The iNOS mRNA was increased in male (**A**) and female (**B**) spontaneously hypertensive rats (SHR) compared to the normotensive rats (WR). Exposure of rats to continuous light for 6 weeks resulted in increased iNOS transcripts in the male SHR, but not in the female. Omacor^®^ treatment suppressed elevated iNOS transcripts in both the male and female SHR rats. Values are the mean ± SD of 6 rats in each group. * *p* < 0.05 compared to WR; # *p* < 0.05 compared to SHR; ♣ *p* < 0.05 compared to SHR-L.

**Figure 8 marinedrugs-19-00659-f008:**
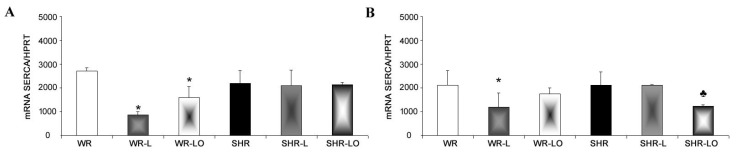
Transcripts of mRNA SERCA were decreased by the 6-week continuous light in male (**A**) and female (**B**) normotensive rats (WR) but not in spontaneously hypertensive rats (SHR). Omacor^®^ attenuated this decreased SERCA transcripts in the male rat and normalised mRNA SERCA level in the female WR. Values are the mean ± SD of 6 rats in each group. * *p* < 0.05 compared to WR; ♣ *p* < 0.05 compared to SHR-L.

**Table 1 marinedrugs-19-00659-t001:** Registered parameters of male and female normotensive rats (WR) and spontaneously hypertensive rats (SHR).

**Male**	**WR**	**WR-L**	**WR-LO**	**SHR**	**SHR-L**	**SHR-LO**
SBP (mmHg)	108 ± 21	129 ± 12 *	117 ± 13	175 ± 6 *	201 ± 19 #	181 ± 12 ♣
HW (mg)	763 ± 74	789 ± 71	801 ± 57	931 ± 107 *	954 ± 81	896 ± 131
HW/BW	2.7 ± 0.1	2.9 ± 0.2 *	2.9 ± 0.2	4.1 ± 0.1 *	4.2 ± 0.1	4 ± 0.2 ♣
LVW (mg)	559 ± 49	566 ± 43	573 ± 41	690 ± 93 *	717 ± 53	679 ± 105
LVW/BW	2 ± 0.1	2.1 ± 0.2	2.1 ± 0.2	3.1 ± 0.1 *	3.2 ± 0.1 #	3.0 ± 0.2 ♣

**Female**	**WR**	**WR-L**	**WR-LO**	**SHR**	**SHR-L**	**SHR-LO**
SBP (mmHg)	109.7 ± 13	123.6 ± 19	106.1 ± 16	154 ± 19 *	170.6 ± 26	150.6 ± 9
HW (mg)	609 ± 23	571 ± 43	586 ± 31	671 ± 87	670 ± 35	622 ± 32 ♣
HW/BW	3.1 ± 0.1	3.2 ± 0.2	3.1 ± 0.1	4.4 ± 0.1 *	4.4 ± 0.1	4.6 ± 0.2 ♣
LVW (mg)	427 ± 24	419 ± 43	431 ± 14	495 ± 57 *	490 ± 25	466 ± 30
LVW/BW	2.2 ± 0.1	2.3 ± 0.2	2.3 ± 0.1	3.3 ± 0.1 *	3.2 ± 0.1	3.5 ± 0.2 ♣

Note that the systolic blood pressure (SBP) of males (up) but not females (down) rats was significantly increased in response to 6-weeks exposure to continuous light (normotensive rats exposed to constant light—WR-L, spontaneously hypertensive rats exposed to constant light—SHR-L). While treatment with Omacor^®^ suppressed the elevation of SBP regardless the sex (normotensive rats exposed to constant light and treated with Omacor^®^—WR-LO, spontaneously hypertensive rats exposed to constant light and treated with Omacor^®^—SHR-LO). Values are the mean ± SD of 12 rats in each group. * *p* < 0.05 compared with WR; # *p* < 0.05 compared with SHR; ♣ *p* < 0.05 compared with SHR-L.

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
