# Peer review of "Omacor Protects Normotensive and Hypertensive Rats Exposed to Continuous Light from Increased Risk to Malignant Cardiac Arrhythmias"

_marinedrugs, 2021, doi:10.3390/md19120659_

Round 1

Reviewer 1 Report

This interesting manuscript describes the how the impact of an environmental stressor that negatively influences cardiac electrical activity is moderated by an omega-3 supplement.   The data presented demonstrate an important range of endpoints, from physiological to molecular, which collectively can begin to provide mechanistic insights into omega-3 PUFA use in the context of cardiac health.      

Major Issues

In the Introduction, the third paragraph (Cx43) could be expanded a bit more to extend into the interrelatedness of  the cardiac electrophysiology and the molecular targets measured.   The current transitions in the Results section from one molecular marker (protein or gene expression) to the next are limited. Providing a rationale for the molecular markers will help readers less versed in cardiac rhythmicity details gain a deeper appreciation for the findings. 

The fourth paragraph of the Introduction suggests/implies that the Omacor product is obtained directly from phytoplankton.  While omega3-enriched phytoplankton are indeed in the food chain of fatty fish, the latter are typically used in the manufacture of omega-3 enriched oils.  Information available about Omacor indicate that it a fish-based product as well.  The triglyceride-containing oils are subsequently processed to the ethyl ester modified EPA and DHA products, such as Omacor.  The authors are encouraged to provide more suitable references than #18.    See for example: https://www.tandfonline.com/doi/pdf/10.2217/17460875.2.3.263

Figure 3: There is no mention of Figure 3 panels E and F in the text or in the legend.  The discussion of this figure is a bit confusing.  For panels C and D, is “total Cx43” a sum of the P0, P1 and P2 bands?  Similarly, for panels E and F, is “P Cx43” a sum of the P1 and P2 bands?   Lines 121-134

In the Discussion (around Line 224-226), the authors are encouraged to state that the absence of measured circulating (plasma/serum) fatty acid levels as a limitation of the study.   The authors are also encouraged to collect and analyze circulating and red cells fatty acid levels in future supplementation studies.  The former serves to verify bioavailability and supplementation compliance and the latter could be used to estimate an omega-3 index. Fatty acid analysis is a relatively simple process of generating fatty acid methyl esters for detection and quantification by gas chromatography with flame ionization detection (GC-FID). 

Minor Issues  

More information in the Methods section regarding how the omega-3 product was provided/administered to animals would be helpful.  Mixed with food, by gavage, or other?

Were animals housed individually? If not, could group housing contribute to another stressor(s) that might contribute to observed gender differences in endpoints measured?

Consistency of data format in Table 1:  Female blood pressure values are shown with a comma. All other non-integer data in this table are shown with a decimal point.  

Etylester should be ethyl ester.

In the Discussion (Line 217-220), gender differences in Cx43 expression are referenced as being greater in females than males.    In Figure 2, male gene expression appears to be greater (by eye) than that in females and there is no apparent gender-based difference in protein levels (Figure 3).  The authors are encouraged to present gender –based differences in tabular form or in the text with the appropriate statistical analyses.   It is not clear, based on the data presented, how the results of the current study agree the cited study. Could Cx43 normalization to GAPDH (current study) vs myosin (ref22) be an issue here?  

In the Discussion, it would be helpful to the reader if Figure or Table references (in the text) to data/observations from the study could be included parenthetically.   

Author Response

We would like to thank you very much for your time to review our manuscript and we appreciate your remarks as well as suggestions aimed to improve quality of the paper. We hope that our responses are relevant to your comments along with their editing (highlighted in brown) in revised manuscript.

Major Issues

In the Introduction, the third paragraph (Cx43) could be expanded a bit more to extend into the interrelatedness of the cardiac electrophysiology and the molecular targets measured.  

Accordingly, we included followed information in revised version.

Noteworthy, the pro-inflammatory factors, such as NF-κB (the nuclear factor kappa-light-chain-enhancer of activated B cells) and iNOS (inducible nitric oxide synthase) have important impact on this process (Baum 2012, Kirca 2015, Andelova 2021). While abnormal calcium handling and calcium overload due to disorders in SERCA2 (sarcoplasmic reticulum Ca2+ATPase) promotes cell-to-cell electrical uncoupling at the Cx43 channels as well as ectopic action potential generation (Landstrom 2017, Andelova 2021). 

The current transitions in the Results section from one molecular marker (protein or gene expression) to the next are limited. Providing a rationale for the molecular markers will help readers less versed in cardiac rhythmicity details gain a deeper appreciation for the findings. 

We revised Results section (see please highlighted by braun colour) and included further information about pro-arrhythmic potential of examined molecular markers.

The fourth paragraph of the Introduction suggests/implies that the Omacor product is obtained directly from phytoplankton.  While omega3-enriched phytoplankton are indeed in the food chain of fatty fish, the latter are typically used in the manufacture of omega-3 enriched oils.  Information available about Omacor indicate that it a fish-based product as well.  The triglyceride-containing oils are subsequently processed to the ethyl ester modified EPA and DHA products, such as Omacor.  The authors are encouraged to provide more suitable references than #18.    See for example: https://www.tandfonline.com/doi/pdf/10.2217/17460875.2.3.263

Yes, we agree with you although we really do not know the details in technology. We added following references Bhatnagar 2007, Kar 2011 as was suggested.

Figure 3: There is no mention of Figure 3 panels E and F in the text or in the legend.  The discussion of this figure is a bit confusing.  For panels C and D, is “total Cx43” a sum of the P0, P1 and P2 bands?  Similarly, for panels E and F, is “P Cx43” a sum of the P1 and P2 bands?   Lines 121-134

We followed your suggestions and revised the figure 3 legend and the text, hopping that it is more clear for readers.

In the Discussion (around Line 224-226), the authors are encouraged to state that the absence of measured circulating (plasma/serum) fatty acid levels as a limitation of the study.   The authors are also encouraged to collect and analyze circulating and red cells fatty acid levels in future supplementation studies.  The former serves to verify bioavailability and supplementation compliance and the latter could be used to estimate an omega-3 index. Fatty acid analysis is a relatively simple process of generating fatty acid methyl esters for detection and quantification by gas chromatography with flame ionization detection (GC-FID). 

Yes, we noted that we did not measure plasma fatty acids levels in current study but we included our previous study (Bacova 2013) showing that omega-3-index was lower in both male and female SHR and increased after omega-3 intake. In this study we measured by gas chromatography in addition to eicosapentaenoic acid and docosahexaenoic acid also linoleic acid, arachidonic acid and alfa-linolenic acid.

Minor Issues  

More information in the Methods section regarding how the omega-3 product was provided/administered to animals would be helpful.  Mixed with food, by gavage, or other?

We consider gavage as a stressful even if skilfully performed, therefore, according to our experience we fed each rat directly with small biscuit soaked with exact dose of Omacor oil. They like Omacor very much J

Were animals housed individually? If not, could group housing contribute to another stressor(s) that might contribute to observed gender differences in endpoints measured?

As we know from literature, rats are social animals and for them is stressful to be housed individually. Thus, in agreement with the approval of our study by the State Veterinary Administration of the Slovak Republic, we housed three rats per cage of size 590 x 380 x 200 mm.

Consistency of data format in Table 1:  Female blood pressure values are shown with a comma. All other non-integer data in this table are shown with a decimal point.  

Etylester should be ethyl ester.

We apologize very much for the mistakes, despite of all authors have read the manuscript L.

In the Discussion (Line 217-220), gender differences in Cx43 expression are referenced as being greater in females than males. In Figure 2, male gene expression appears to be greater (by eye) than that in females and there is no apparent gender-based difference in protein levels (Figure 3). The authors are encouraged to present gender –based differences in tabular form or in the text with the appropriate statistical analyses. It is not clear, based on the data presented, how the results of the current study agree the cited study. Could Cx43 normalization to GAPDH (current study) vs myosin (ref22) be an issue here?  

We agree that in figure 2 the gene expression of GJP43 gene transcripts in females (2B) is lower than in males (2A). The reason for different scale in A and B graph should be that the samples and PCR analyses for males and females were performed in different days. Therefore, we compared the changes of GJP43 gene transcripts to control males and to control females. However, the trends of changes among examined groups differ between males and females.

We would like to note that in the current study we did not perform western bot analysis of males and females in one gel/membrane as it was done in reference 26 (in revised version). Since we loaded 13 samples from male rats plus protein marker (the same for female) we did it separately, because maximum 15 samples can be loaded on one gel/membrane. Therefore, we cannot compare real values of proteins between males and females. We can only compare whether alterations of assessed proteins differ or not in males versus controls and females versus controls.  For example, comparing to control WR Cx43 protein levels was not affected by continuous light in males but increased in females. It suggests different gender related response.

Normalisation to “house keeper” is used to avoid loading (pipetting) discrepancies of protein samples.  Normalisation to “house keeper” (GAPDH, myosin) did not affect the relative value of assessed protein.

In the Discussion, it would be helpful to the reader if Figure or Table references (in the text) to data/observations from the study could be included parenthetically.   

Yes, we included the numbers in the text referring to the Figures in Discussion section.

Thanks for your help to increase the chance to publish our study.

Reviewer 2 Report

The authors presented a systematic approach in demonstrating the role of high omega 3 intervention (Omacor) and molecular mechanisms in preventing ventricular fibrillations in a certain group of experimental rats.

In light of a recent clinical study showing some increase in atrial fibrillation with increased omega 3 (not Omacor) as potential risk (ref: JAMA. 2021;325(11):1061-1073. doi:10.1001/jama.2021.1489), the authors might want to address this apparent contradiction in their discussion as implications of these results and results from their study.

In Figs 3A/B, 4A/B, 5A/B, I noticed that the immunoblots have been spliced.  This might be due to satisfy space requirements and facilitate comparison.  I assume that the spliced blots are from the same gel and just adjusted to make it closer for better comparison. Is that right?  The authors might want to clarify this in the Materials and Methods section to prevent the reader from having any doubts.

Other minor edit such as spell-out SERCA in the abstract for consistency.

Author Response

We would like to thank you very much for your time to review our manuscript and we appreciate your remarks and suggestions aimed to improve quality of the paper. We hope that our responses and revision of the manuscript reflect your comments.

In light of a recent clinical study showing some increase in atrial fibrillation with increased omega 3 (not Omacor) as potential risk (ref: JAMA. 2021;325(11):1061-1073. doi:10.1001/jama.2021.1489), the authors might want to address this apparent contradiction in their discussion as implications of these results and results from their study.

We have included this most recent trial (Albert et al. 2021) in revised manuscript and discussed that considering very heterogeneous population (see please Table 1) of older participants receiving 840 mg/d EPA-DHA (like 1 capsule of Omacor), it is not surprising that there was not significant difference in the risk of incident AF over 5 years of follow-up. As we have emphasised previously (Tribulova et al 2017) the major flaw of clinical studies (including cited study) is luck of information about basal omega-3 status of participants/patients. Accordingly, the relevance of the conclusions of clinical trials is difficult to justify.  

In Figs 3A/B, 4A/B, 5A/B, I noticed that the immunoblots have been spliced. This might be due to satisfy space requirements and facilitate comparison.  I assume that the spliced blots are from the same gel and just adjusted to make it closer for better comparison. Is that right?  The authors might want to clarify this in the Materials and Methods section to prevent the reader from having any doubts.

Yes, you are right. We clarified that the representative immunoblots (proteins and GAPDH) in figures was spliced to allow the comparison. The spliced blots were from the same gel to prevent the reader from having any doubts.

Other minor edit such as spell-out SERCA in the abstract for consistency.

We spelled-out sarcoplasmic reticulum Ca2+ATPase in the abstract.

Thanks for your support to publish our study.